# Psychological Traditionality and Modernity and Happiness: The Different Happiness Levels in Chinese Students

**DOI:** 10.3390/bs14040304

**Published:** 2024-04-08

**Authors:** Erjia Xu, Sihua Liao, Ping Hu

**Affiliations:** Department of Psychology, Renmin University of China, Beijing 100087, China; erjia@ruc.edu.cn (E.X.); sihualiao@ruc.edu.cn (S.L.)

**Keywords:** happiness, Chinese psychological traditionality, Chinese psychological modernity, individual-level happiness, relational-level happiness, societal-level happiness

## Abstract

The conceptualization of happiness varies across different cultures. In Chinese culture, happiness includes oneself and hinges on others. Chinese social development has influenced psychological traditionality (PT), psychological modernity (PM), and personal happiness. Our study recruited 450 participants to examine the different happiness levels in Chinese students with diverse PT and PM. The results indicate that individuals scoring higher in PT and PM reported higher life satisfaction. Moreover, individuals scoring higher in PT reported more positive emotions, fewer negative emotions, and greater social well-being, while those scoring higher in PM reported more negative emotions and greater relationship happiness. The happiness of Chinese students comprised individual, relational, and societal levels and happiness at different levels related to Chinese PT and PM. The present study may promote cross-cultural understanding and potentially inform interventions for individual happiness within positive psychology.

## 1. Introduction

“Happiness depends upon ourselves.”—Aristotle.

Happiness has been a pursuit of humanity since ancient times. Since the 1960s, numerous disciplines have conducted research into “happiness”. In particular, in psychology, from a basis in disciplinary characteristics, research has focused on paving the way for people to achieve happiness and on the impact of a sense of happiness on individuals’ psychology and behavior [1].

Although there has been considerable research into the sense of happiness [2], numerous issues still permeate happiness studies. Of these, cross-cultural consistency is paramount. Following the cultural revolution in psychology in the 1980s, researchers noticed that the same psychological terminology has different connotations in different cultures. As early as the beginning of the 21st century, researchers identified differences in the concept of happiness in Chinese and American culture from interviews with college students in Taiwan and the United States [3]. Not only does the word “happiness” differ in its written form in different cultures, but its connotations are also varied, and within this variation, the concept of happiness in Chinese culture is particularly noteworthy. Researchers found that, in 30 cultures, 24 countries associate happiness with “luck”, and Chinese culture’s interpretation of happiness had its own characteristics [4]. Therefore, investigating the constituents and connotations of happiness within Chinese culture can, from a theoretical standpoint, enrich theories of happiness and lay the groundwork for future research. From a practical viewpoint, it can also enhance other cultures’ comprehension of the state of happiness within Chinese culture, thereby promoting cross-cultural communication and exchange. From the perspective of positive psychology, comprehending what constitutes happiness enables the implementation of interventions to enhance individual happiness.

### 1.1. What Is Happiness

Happiness is a positive inner experience and, as the highest good and the ultimate motivator for all human behaviors, has attracted ever-increasing attention from psychologists over past decades [5,6]. In numerous cross-cultural studies, people in Eastern societies, such as China, have tended to have lower scores for happiness. Many researchers have attributed this to their traditional culture, which is rooted in collectivism, believing that individuals within cultures favoring individualism were expected to experience higher levels of happiness [7]. Based on this perspective, the World Happiness Report [8] indicated that subjective well-being should be measured based on a tripartite framework featuring life evaluations and positive and negative emotions.

However, the view that happiness can be universally defined and is experienced in the same way across different cultures may need to be revised. The definition and measurement of “happiness” vary among different cultures. The connotations of happiness for Chinese individuals diverge from a Western perspective; happiness for Eastern people is often considered a dialectical balance. Blessings and misfortunes are interdependent and can transform into each other, suggesting that instead of pursuing joy to excess, one should seek inner peace and harmony with the outside world [9]. For the Chinese, happiness extends beyond hedonistic sensory pleasure. It is achieved through harmonious co-existence with others, society, and nature, leading to inner tranquility [10]. Consequently, the Chinese concept of happiness encompasses an individual’s satisfaction with their life and comprehensive evaluations of relationships, family, and society to reach a state of harmony.

Therefore, to investigate happiness among Chinese individuals, one must employ multiple measurement tools to evaluate from various angles and dimensions. A classic tool for measuring happiness is the Satisfaction with Life Scale (SWLS) [11], which assesses general life satisfaction. The assessment of happiness should also incorporate the measurement of individual emotions. Therefore, we also adopted the Positive and Negative Affect Schedule scale (PANAS) [12,13]. The measurement method that combines SWLS with PANAS is consistent with the previously used one for subjective well-being [5]. The previously mentioned scales for measuring happiness—the SWLS or the PANAS—all gauge happiness from the perspectives of quality of life and individual subjective experiences. This research orientation is anchored in the philosophical foundation of hedonism [14]. However, as mentioned earlier, the happiness of Chinese individuals encompasses not only personal contentment but also the assessment of social relationships. In other words, happiness is not solely the attainment of individual pleasure but includes the perfect experience achieved by establishing good relationships with others. Moreover, Ryff has utilized the term “psychological well-being” to describe this subjective happiness [15]. In this context, the relational dimension of happiness could reflect the relational characteristics of Chinese individuals’ sense of happiness.

Furthermore, the happiness of family members holds significant importance for Chinese individuals and has served as one of the crucial criteria for determining their happiness [16]. Therefore, to comprehensively measure the happiness of Chinese individuals, it is essential to incorporate the happiness of their family members as an integral component of overall happiness. Further expanding upon the relational dimension of psychological well-being, social well-being is an individual’s self-judgment and evaluation of their relationships with others, the community, and society. Social well-being explores a person’s well-being in a broader social context [17,18].

To summarize, this study explores the concept of happiness by considering its various levels. It encompasses individuals’ cognitive evaluations of their life satisfaction based on their standards (measured by scales such as the SWLS and PANAS), the positive experiences derived from establishing fulfilling relationships with others (as reflected in the psychological well-being dimension of relationships and the happiness of family members), as well as the overall well-being achieved from a societal perspective (social well-being). The following sections will further examine the importance of relationship happiness and social well-being in the happiness of Chinese individuals.

### 1.2. The Different Happiness Levels in Chinese Culture

Previous research has found that people from collectivist cultures are more willing to sacrifice their desires and conform to the group’s will [19]. In such cultures, there is a more vital link between an individual’s perception of cultural norms and life satisfaction. Compared with Westerners, research has found six unique sources of happiness for Chinese people: harmonious relationships with friends and family, praise of others, better life conditions than others, acceptance of fate, material satisfaction, and work achievements [20]. The first three items are closely related to relationships with others. Subsequent research has validated the causal influence of social support and expectations on happiness [21].

In summary, based on the philosophical roots of Buddhism and Confucianism, the happiness of others and society not only provides a reference standard for individuals to judge their happiness but is also interdependently connected to personal happiness. As individuals help others achieve happiness, they can actualize their happiness [22]. In other words, under the influence of traditional Chinese thought, the happiness of Chinese people is intimately linked to others and society. Researchers have posited that, in addition to their happiness, Chinese people’s concept of happiness also includes concern for the happiness of others. In other words, for Chinese people, feeling good on one’s own does not constitute happiness. Having good relationships with others and seeing their loved ones happy can form Chinese people’s view of happiness. Buddhism, which occupies an important position in Chinese culture, posits that while a person strives to become a Buddha, they should also save all sentient beings from suffering, as true liberation is achieved among the multitude. Moreover, Buddhism also insists that even after enlightenment, one should try to help others reach a state of bliss [23]. Buddhist philosophy asserts that our happiness can only be achieved through the happiness of others [24]. This perspective does not advocate for individuals to neglect their happiness, but rather encourages individuals to do their utmost to alleviate the suffering of others and contribute to their long-term welfare. This approach is considered a pathway to personal happiness. Buddhist thought contends that genuine individual happiness arises from the earnest desire to hope that everyone else can have a meaningful life [25].

Similarly, the primary ideological source of Chinese culture comes from Confucian thinking, which advocates the pursuit of personal happiness and the happiness of all people. It requires individuals to benefit everyone beyond just benefiting themselves [26]. Confucianism suggests that individual happiness stems from moral cultivation and involves some control over material aspects. At the same time, it simultaneously advocates for a life development theory involving “cultivating morality, regulating family, ordering the state, and peace will prevail throughout the universe”, thus realizing the societal happiness of a unified world [27]. Specifically, Confucian thought first affirms the material element of happiness, placing it within the primary context of survival and living, and secondly promotes the union and integration of personal and societal happiness. Individuals should strive to maximize their capacity to attain personal happiness and consequently make significant contributions to the cultural development of society, thereby achieving greater happiness. Therefore, under the influence of traditional Chinese thought, the concept of happiness in China encompasses individual, relationship, and societal happiness.

### 1.3. How Chinese Cultural Values Affect Happiness

Under the influence of traditional Chinese thought, Chinese individuals may emphasize the happiness of others and society. Since the reforms and opening-up policy of the 1980s, Chinese society has transitioned from a traditional to a modern society. Consequently, there has been a gradual transformation in the mindset of Chinese individuals. Yang [28] proposed that through the process of social change, individuals in China experienced changes in their values, ideological forms, and lifestyle habits. During the process of transformation, Chinese individuals manifested two distinct cultural value orientations: psychological traditionality (PT) and psychological modernity (PM). PT reflects individuals’ cognitive attitudes, ideological concepts, value orientations, temperament traits, and behavioral intentions that align with traditional Chinese societal norms. Conversely, PM represents individuals’ cognitive attitudes, ideological concepts, value orientations, temperament traits, and behavioral intentions that are congruent with contemporary Chinese societal norms. The two have been called “modal personality”, which varies as a concept across socio-cultural contexts. In previous research, the individual multidimensional traditionality and modernity scale (IMTMS) [29] has been used to measure the cultural values of individuals related to PT and PM. Specifically, the measurement of PT has included five aspects: middle-way mentality, fatalistic superstition, male superiority, relationship orientation, and filial piety and respect for elders. The measurement of PM has included fairness and justice, consumption orientation, internal locus of control, the quest for novelty and change, and independence and autonomy (detailed questions can be found in Section 2.5, Measurement). It is worth noting that PT and PM have a complex relationship, rather than a simple correspondence or an inverse relationship. PT and PM are not completely contradictory [28]. Instead, individuals often possessed traditional and modern attributes [30]. Chinese society as a whole has functioned in a state of coexistence between tradition and modernity. Therefore, most Chinese individuals have simultaneously held traditional psychological and behavioral attributes while adapting to modern life, indicating that they may possess both PT and PM in their cultural values [31].

Previous research has explored the relationships between the self [32], morality [33], and individual cultural values. However, previous studies on the relationship between cultural values and happiness have typically focused on either PT or PM [34,35] or only measured a single indicator of happiness [36]. However, the happiness of Chinese individuals encompasses not only individual happiness but also relational happiness and societal happiness. Previous studies have mostly found a more substantial positive predictive effect of PT on happiness [37,38]. However, research has also found a positive correlation between PT and anxiety among college students [39], which suggests that PT may not necessarily have a positive association with happiness. Moreover, most studies have yet to look at the potential role of PM. Researchers have argued that happiness may be a casualty of modernity in Western societies [40]. However, other studies found that the increased opportunities for successful modernization contributed to greater subjective happiness among individuals in Germany [41]. It should be noted that China’s modernity is more than just a complete Westernization [42]. Traditional Chinese culture is increasingly valued in contemporary society, and there has been a trend towards a cultural return. The rationality of traditional culture could compensate for the deficiencies of Western civilization, thus forming a unique modern civilization in Chinese society [43] (p. 298). Research has analyzed data from the Chinese Longitudinal Healthy Longevity Survey (CLHLS) from 1998 to 2018 using OLS regression models, and the findings suggested that marketization (China’s modernization) lowers the level of psychological well-being among elderly Chinese populations [44]. However, other research has found a gradual improvement in the mental health of less well-off college students from 1998 to 2015 [45]. These inconsistent results suggest that PM may be positively correlated with negative emotions, but the level of individual happiness among Chinese people may gradually decline during societal modernization unless individuals can benefit from the process. However, no direct research is currently examining the relationship between PM and happiness. Thus, it is only possible to speculate about the relationship between PM and individual happiness.

PT and PM may also relate to individuals’ sense of happiness at both relational and societal levels. Previous research has found that employees with higher PT are more likely to be influenced by the emotions of their leaders [46]. Chinese individuals derive their subjective well-being from positive emotional exchanges with others, and individuals with higher levels of PT tend to place a stronger emphasis on interpersonal relationships and the happiness of others [47]. Under the influence of Chinese traditional thought, the importance placed on others also affects individuals’ social behaviors. Previous studies have found that Chinese people with higher PT exhibited fewer environmentally damaging behaviors due to higher levels of social responsibility [48]. Consequently, individuals with higher levels of PT may experience greater levels of relational happiness and societal happiness. Continuing from the perspective of social change, we discuss the potential relationship of PM with the happiness circle. Using large-scale online survey data from China, it was found that individuals who were younger, single-child, financially wealthy, and living in urban areas demonstrated higher levels of self-focus than those who were older, non-single-child, less financially endowed, and living in non-urban areas [49]. The former also showed higher levels of psychological modernity [50]. Therefore, individuals with higher levels of PT may experience greater individual happiness.

A review of the literature suggests that happiness among Chinese individuals has been examined as satisfaction at the individual level and that it must also be examined at the relational and societal levels. To our knowledge, the present study is the first to examine the relationships between PT and PM with happiness measured at different levels, and therefore offers a new conceptual framework for the study of happiness. Furthermore, studying the different levels of happiness can assist individuals in cultivating positive psychological states by understanding the constituents of an individual’s happiness, more effectively enhancing their happiness and fostering a healthy mindset.

The current study attempted to answer the following question: What are the relationships between PT and PM and different levels of happiness among Chinese students?

The current study measured the sense of happiness of Chinese participants at three levels: the individual, relational, and societal levels. Based on instruction, the following hypotheses are proposed:

**Hypothesis** **1:***At the individual level, individuals who score higher in PT may report higher life satisfaction and a higher frequency of positive emotional experiences. Conversely, individuals who score higher in PM may report lower life satisfaction and a higher frequency of negative emotional experiences*.

**Hypothesis** **2:***At the relational level, individuals who score higher in PT may report higher familial happiness and relationship happiness. In contrast, individuals who score higher in PM may report lower familial happiness and relationship happiness*.

**Hypothesis** **3:***At the societal level, individuals who score higher in PT may report higher social happiness. Conversely, individuals who score higher in PM may report lower social happiness*.

## 2. Methods

### 2.1. Design

The current study utilized a questionnaire-based approach to investigate the research question, and the individual multidimensional traditionality and modernity scale to measure PT and PM. Additionally, separate questionnaires were employed to measure individual-level, relational-level, and societal-level happiness. The questionnaires used in the present study are detailed in Section 2.5, Measurement.

### 2.2. Setting

Participants completed the questionnaire via a survey platform (https://www.wjx.cn/) in exchange for a participation fee. All participants volunteered for this study and signed an informed consent form. The recruitment of participants and the collection of questionnaires were executed from 28 July to 31 July 2021.

Participants first provided basic demographic information, including gender, age, subjective social class, and objective social class as control variables, following previous research [51]. Then, participants read the questionnaire instructions and signed an informed consent form. Once they began the primary survey, participants were requested to complete all questionnaires consecutively and in a quiet environment. The questionnaires were presented randomly. The entire questionnaire was designed to be completed within approximately 20 min. The questionnaire included random general knowledge questions (e.g., “Is the average temperature higher in summer than in winter”) and tests that required the participant to select the corresponding option (e.g., for this question please choose the fourth option). The questionnaire included five screening questions for careless responding. Participants who answered any of the screening questions incorrectly would be excluded from the data analysis, and those whose date were not excluded automatically received CNY 20.

### 2.3. Participants

Posters were put up in major universities in Beijing, and links to the online questionnaire were posted in recruitment groups to solicit respondent’ participation. Participants were required to be non-psychology major students aged between 18 and 30 years old, and capable of participating and completing the questionnaire in a timely and conscientious manner. Four hundred and fifty participants were involved in this study.

### 2.4. Variables

Psychological traditionality (PT) refers to individuals possessing cognitive attitudes, ideological concepts, value orientations, temperament traits, and behavioral intentions that are consistent with traditional Chinese society.

Psychological modernity (PM) represents individuals having cognitive attitudes, ideological concepts, value orientations, temperament traits, and behavioral intentions that align with modern Chinese society.

Individual-level happiness refers to the evaluation of one’s own sense of happiness at the individual level, encompassing personal life satisfaction as well as experiences of positive and negative emotions in life.

Relational-level happiness refers to individuals’ evaluation of happiness at the relational level, which includes individuals’ perceptions of their family members’ happiness as well as the quality of their relationships with others.

Societal-level happiness refers to individuals’ evaluation of happiness at the societal level, which includes the self-assessment of relationships and their quality among individuals, groups, and society as a whole.

### 2.5. Measurement

**Individual Multidimensional Traditionality and Modernity Scale (IMTMS)** [29]. The scale consisted of two sub-scales, PT and PM, rated on a 7-point Likert scale ranging from 1 (completely disagree) to 7 (completely agree). Specifically, the PT sub-scale included 40 items, while the PM sub-scale included 36 items. Each sub-scale includes five dimensions. The traditionality sub-scale comprises the following dimensions: middle-way mentality (e.g., individuals should constantly examine their behaviors and correct their mistakes), fatalistic superstition (e.g., the success or failure of every individual is predestined), male superiority (e.g., men are the heads of the family and should make decisions regarding family matters), relationship orientation (e.g., before making decisions, one should consider the opinions of others), and filial piety and respect for elders (e.g., younger generations should show respect and obedience to their older relatives). The modernity sub-scale includes the following dimensions: fairness and justice (e.g., basic human rights should be protected in a democratic society), consumption orientation (e.g., life should include enjoying good food, clothing, and housing), internal locus of control (e.g., no matter how unfavorable the environment, as long as a person works hard and perseveres, they will eventually succeed), quest for novelty and change (e.g., people should have diversified interests and be willing to learn new things), and independence and autonomy (e.g., people of different religious beliefs can still be friends). The various dimensions were utilized to measure different aspects of PT and PM. However, the current study did not explore the relationship between the content of PT or PM and its association with happiness. Therefore, the total scores of the two sub-scales represented the levels of PT and PM. Higher scores indicated a greater identification with these cultural values. In the current study, Cronbach’s alpha coefficients for PT and PM were 0.943 and 0.874, respectively.

At the individual level, the measurement of happiness included two scales:

**Satisfaction With Life Scale (SWLS)** [52]. This scale consisted of five items rated on a 7-point scale. An example item is “I am delighted with my life”, with participants requested to express their agreement or disagreement, from 1 (strongly disagree) to 7 (strongly agree). A higher total score indicated greater overall satisfaction with life. Cronbach’s alpha coefficient for SWLS was 0.909 in this study.

**Positive and Negative Affect Schedule (PANAS)** [53]. This scale utilized a 5-point scoring system, from 1 (representing rare or non-existent) to 5 (representing very frequent). The Positive Affect sub-scale (PANAS_P) and Negative Affect sub-scale (PANAS_N) each comprised ten items. The scale was used to evaluate the frequency of experiencing positive and negative emotions daily. A higher score indicated a higher frequency of feeling the respective emotion. In this study, Cronbach’s alpha coefficients for PANAS_P and PANAS_N were 0.902 and 0.921, respectively.

At the relationship level, the measurement of happiness included two scales:

**Satisfaction With Family Scale (SWFS)**. This scale was an adaptation of the SWLS specifically designed to measure an individual’s perception of their family members’ satisfaction with life (e.g., my families are delighted with their life). Like SWLS, the scale consisted of five items rated on a 7-point scale (from 1—strongly disagree to 7—strongly agree). A higher total score indicated more excellent individual families’ satisfaction with life. Cronbach’s alpha coefficient for SWFS was 0.932 in this study.

**Psychological Well-Being Scale (PWB)** [15]. This scale assessed various dimensions related to self. The dimension of “Positive Relationships” (PWB_PR) was measured in this study (e.g., many people consider me to be a warm and loving person). This scale consisted of 14 items and utilized a 6-point scoring system, from 1 (strongly disagree) to 5 (strongly agree). Higher scores on this dimension indicated that an individual has warm, satisfying, trusting relationships with others, is concerned about the welfare of others, is capable of intense empathy, affection, and intimacy, and understands the give and take of human relationships. Cronbach’s alpha coefficient for PWB_PR was 0.769 in this study.

At the societal level, the measurement of happiness included one scale:

**Social Well-Being Scale (SWBS)** [17]. This scale consisted of five dimensions: social acceptance, social actualization, social contribution, social coherence, and social integration. In total, 15 items were rated on a 7-point scale, from 1 (strongly disagree) to 7 (strongly agree). The version of the SWBS scale used in this study does not involve the need for reverse scoring of any items. Additionally, it is essential to note that this study did not investigate the relationship between PT and PM and different aspects of social well-being. Therefore, the total score was used to indicate individuals’ perceived social well-being in this study. The higher total scores indicate higher levels of social well-being. An example item was “I feel close to others”. Cronbach’s alpha coefficient for SWBS was 0.913 in this study.

**Control Variables**: The measurement of gender, age, and objective social class followed past research [51], requiring participants to directly report their gender (selecting from male/female), their age (fill in the blank), and their family annual income (fill in the blank). The measurement of subjective social class (SSC) utilized the Subjective Social Status Scale [54]. This scale provides participants with a 10-level ladder illustration, asking them to envision where they stand on this ladder. The ladder represents each individual’s perceived social class, with scores closer to 10 indicating a higher social standing. A score of 1 indicates the lowermost layer of the social class, while 10 signifies the uppermost layer.

### 2.6. Bias and Study Size

This study measured all variables through self-report measures, which may have introduced common method bias. Several measures were taken to effectively control for common method bias. Firstly, the questionnaire instructions emphasized the anonymity and confidentiality of the data, and participants were requested to respond based on their actual circumstances. Secondly, different scales were adopted as much as possible to minimize common method biases, each with varying rating metrics.

Using G*Power 3.1 for a priori power estimation [55], F tests were selected to conduct a multiple linear regression analysis with three tested predictors and a total of six predictors. The effect size, *f*^2^, was set at 0.15, with a significance level of 0.05 and a desired power of 95%. The results indicate that 119 participants were required. Thus, the sample size for this study is deemed sufficient.

### 2.7. Statistical Methods

The data were analyzed using SPSS 25.0. Since the measurement scales for different aspects of happiness assess specific facets of happiness, it was not appropriate to combine the results from different scales. Therefore, the subsequent analyses of this study separately explored the relationships of PT and PM with different levels of happiness across various facets. Firstly, data were screened based on the participants’ completion of the questionnaire and their responses to the screening questions. Next, a Pearson correlation analysis was carried out on the main variables. Finally, using hierarchal regression analysis, the participant’ gender, age, objective social class, and subjective social class were set as control variables. This analysis examined the relations among predictor variables PT and PM and the criterion variables at different levels of happiness, thereby testing Hypotheses 1, 2, and 3.

## 3. Results

Out of the initial sample of 450 participants, 32 individuals did not complete the questionnaires or pass the screening questions. As a result, 418 participants were included in the subsequent analysis. The descriptive statistical results of the participant’s demographic information can be found in Table 1.

### 3.1. Controlling and Testing for Common Method Bias

The Harman single-factor test was conducted to examine the presence of common method bias in the data [56]. Exploratory factor analysis was performed on all variables, resulting in 10 factors with eigenvalues greater than 1. The variance explained by the first factor was only 28.26%, which is significantly below the critical threshold of 40%. Therefore, this survey has no apparent issue of common method bias.

### 3.2. Descriptive Statistics and Related Analysis

The correlations between PT and PM (treated as continuous variables) and different levels of happiness were examined. The Pearson correlation coefficients were calculated, and the results are presented in Table 2. Due to the inclusion of eight sets of correlations in this study, a Bonferroni adjustment [57] was performed to account for multiple comparisons. The significance level for establishing statistical significance was set at *p* < 0.006 (0.05 divided by 8). The findings indicated that PT positively correlated with PM. At individual-level happiness, PT positively correlated significantly with PANAS_P, and PM positively correlated significantly with SWLS and PANAS_P. At relational-level happiness, only PM positively correlated significantly with PWB_PR. At societal-level happiness, PT positively correlated significantly with SWBS.

### 3.3. Hierarchical Regression Analysis

Based on these results, hierarchical regression analyses were conducted, with individual-level happiness, relationship-level happiness, and societal-level happiness as criterion variables. PT and PM were used as predictor variables. Demographic variables (age, gender, subjective and objective social class) were treated as control variables. In the analysis process, Model 1 included only the control variables as predictor variables, with different levels of happiness as the criterion variables. Model 2 incorporated the predictor variables of PT and PM while still using different levels of happiness as the criterion variables. Both non-adjusted and adjusted *R*^2^ values are used to explain the goodness of fit of the model. An examination of collinearity indicated VIF values between 1.10 and 1.52, suggesting the absence of multicollinearity.

The results of the hierarchical regression analysis on individual-level happiness are presented in Table 3. Results showed a significant positive predictive role of PT on SWLS and PANAS_P while a significant negative predictive role of PT on PANAS_N. Additionally, PM demonstrated a significant positive predictive role on SWFS and PANAS_N. With controlling for relevant variables, the inclusion of PT and PM in the regression models led to increased variance explained in the criterion variables. Specifically, the inclusion of PT and PM accounted for an additional 7% in variance explained for SWLS, 9.9% for PANAS_P, and 10% for PANAS_N.

The hierarchical regression analysis results on relational-level happiness (see Table 4) showed that PT significantly and positively predicted SWFS, while PM significantly and positively predicted PWB_PR. With controlling for relevant variables, the inclusion of both PT and PM in the analysis further enhanced the explanatory power of the criterion variables. Specifically, the inclusion of the PT and PM variables increased the explanatory power by 4.8% for SWFS, while the explanatory power for PWB_PR was increased by 12%.

The hierarchical regression analysis results on societal-level happiness (see Table 5) showed that PT significantly positively predicted SWBS, while PM did not significantly predict SWBS. With the controlling for relevant variables, the inclusion of both PT and PM significantly improved the explanatory power of the criterion variable SWBS. The additional contribution of PT and PM resulted in a 13.9% increase in variance explanatory power for SWBS.

## 4. Discussion

In this study, a questionnaire method was employed to collect data on psychological traditionality (PT) and psychological modernity (PM). Additionally, participants were asked to report their levels of happiness at the individual, relational, and societal levels to verify Hypotheses 1, 2, and 3. Firstly, a correlation analysis revealed a significant positive correlation between PT and PM (*r* = 0.346, *p* < 0.001), which aligned with previous research findings [58]. This suggested that Chinese students exhibited both traditional and modern cultural values, and that their psychological and behavioral patterns may be influenced by both value systems simultaneously. Consequently, it was necessary to analyze PT and PM separately in relation to different levels of happiness rather than treating them as a single variable offsetting the other.

For individual-level happiness, the correlation results (Table 2) showed that PT positively correlates significantly with PANAS_P, and PM positively correlates significantly with SWLS and PANAS_P. The results of the hierarchical regression analysis (Table 3) showed that individuals who scored higher in PT and PM both reported higher life satisfaction (SWLS). Individuals who scored higher in PT reported more positive emotions and fewer negative emotions, while individuals who scored higher in PM reported more negative emotions, conversely. The content of the emotional experience portion within H1 has been validated. Furthermore, individuals who scored higher in both PT and PM reported higher levels of life satisfaction. This finding was consistent with previous research [59], indicating that, in contemporary Chinese society, individuals who embrace both traditional and modern cultural values can find cultural identity within this duality, leading to higher levels of life satisfaction. The results of this study found that both PT and PM positively predicted an individual’s happiness. Previous research has found that an individual’s level of cultural fit can predict their happiness [60]. The results of this study have demonstrated that current Chinese society encompasses both traditional and modern characteristics. In the process of global modernization, modern society has received considerable criticism, standing accused of degrading the quality of life and making people less happy. However, comparisons across 141 countries reveal that people living in modernized societies tend to be happier [61]. Modernization may have brought convenience and improved quality of life, which lead to higher life satisfaction. In the current study, the focus was on individual values: PT and PM. The middle-way mentality inherent in PT enables individuals to face life’s challenges more positively, experiencing more positive emotions and higher life satisfaction [62]. Conversely, values such as consumption orientation and independence and autonomy in PM may contribute to increased competition and pressure among Chinese individuals, leading to more negative emotions [63] and lower life satisfaction. The relationship between PT, PM, and emotional experience may be related to the ideal affect [64]. Previous research has found that Chinese students who scored higher in PT tended to experience low-arousal positive emotions. In contrast, those who scored higher in PM tended to experience high-arousal emotions, whether positive or negative [65]. This can explain why Chinese students who scored higher in PM may still report higher levels of life satisfaction despite experiencing negative emotions. Future research could extend the findings of this study by delving deeper into the correlation between various dimensions of PT and PM and individuals’ happiness.

For relational-level happiness, the correlation results (Table 2) showed that PT and PM did not correlate significantly with SWFS, while PM positively correlated significantly with PWB_PR. The results of the hierarchical regression analysis (Table 4) indicated that individuals who scored higher in PT tended to report higher family happiness. In contrast, individuals who scored higher in PM scores tended to report higher scores of positive relationships. This finding partially supports H2, which posits a positive correlation between PT and family happiness. The relation of PT with SWFS confirms previous research findings, demonstrating that individuals with traditional Chinese cultural values tend to have higher family happiness [66]. However, an exciting result emerged, as individuals who scored higher in PM reported more positive relationships. This might be due to the difference in the concept of relationships in modern Chinese society. Chinese PT emphasizes sacrificing one’s interests to fulfill others’ needs, while Chinese PM underlines equal exchanges in relationships [67]. Additionally, in the context of research on the adaptation of Chinese students, it was found that students with lower levels of traditional beliefs may have higher self-efficacy and better adaptation [68]. These findings have contributed to our understanding of the relations between cultural value and relational-level happiness. Previous research has also highlighted the importance of establishing positive relationships with others as a pathway to individual happiness [69]. Indeed, an individual’s social capital and the degree of emphasis they place on social relationships can predict their happiness [70].

For societal-level happiness, consistent results were observed between the correlation (Table 2) and the hierarchical regression analysis (Table 5), indicating that individuals who scored higher in PT tended to report higher levels of social well-being. This finding confirmed the relationship between PT and societal-level happiness, as in H3. However, no significant relationship was found between PM and societal-level happiness. This finding aligned with expectations, as individuals with higher traditional cultural values often prioritize collective sentiments and social harmony [71].

The findings of this study revealed that individuals with a higher adherence to psychological traditionality do not exclusively exhibit other-oriented tendencies but also demonstrate higher levels of self-orientation. This result contradicted previous beliefs that Chinese individuals were predominantly other-oriented [72,73]. Moreover, the conclusion that Chinese individuals were other-oriented has been derived through comparisons with Western cultures in previous studies. Hence, when describing Chinese individuals, it was essential to avoid focusing solely on their other-oriented tendencies and instead to consider both self-orientation and other-orientation within a cross-cultural framework.

To summarize the findings of this study, it was observed that individuals with different degrees of cultural values reported different degrees of happiness at the individual level, relational level, and societal level. Specifically, it appeared that individuals who scored higher in PT reported higher life satisfaction, more positive emotional experiences, fewer negative emotional experiences, higher family satisfaction, and higher social well-being. Individuals who scored higher in PM reported higher life satisfaction, more negative emotional experiences, and more positive relationships with others. The results of Study 1 revealed that PT and PM have different relationships with happiness at different levels. Individuals with different PT and PM scores reported varying levels of happiness at different levels.

This study did not confine itself to the exploration of life satisfaction but expanded research on happiness to the individual, relational, and societal levels. It is crucial to acknowledge that the research methodology employed has its limitations. Firstly, the reliance on subjective reports might introduce bias due to social desirability, as individuals may prioritize the reporting of happiness in intimate relationships and societal happiness. The study also did not adopt experimental methods to explore the causal effects of cultural values on happiness. Moreover, it is worth noting that the participants in this study were predominantly undergraduate and higher education students aged between 20 and 30 years old. Therefore, the generalizability of the study results to other populations is limited. In addition, although this study investigated the components of individual happiness and the levels of the happiness circle within a Chinese cultural context, it still employed happiness scales developed from a Western cultural background. These were operationalized to fit the Chinese conception of happiness without an investigation of a happiness circle originating from the perspectives of Chinese indigenous culture.

Based on the above limitations, future research on happiness may consider the following directions. Firstly, researchers should attempt to expand the age, gender, and educational level ranges of the participant population to examine the characteristics of the happiness circle in non-student samples. The sample could be more representative in terms of demographic variables such as gender, age, education level, and income range. This would enable researchers to assess the sociodemographic differences in happiness and provide a more comprehensive understanding of the composition of happiness. Secondly, a comparative approach can be adopted by investigating participant groups from other countries and sociocultural backgrounds in order to compare the results of this study. This has the potential to uncover cross-cultural similarities and differences in happiness, further elucidating the effect of culture on happiness. Lastly, as mentioned in the introduction, psychological traditionality and psychological modernity are not exclusive values of Chinese culture. Future researchers may consider employing cultural priming methods to explore the effect of cultural values on happiness further, possibly contributing an answer to the ultimate question of how to make humanity happier.

## 5. Conclusions

In summary, this study revealed the relationship between Chinese psychological traditionality and modernity, as well as happiness, and examined the relations of cultural values to different levels of happiness among Chinese students.

At the individual level of happiness, Chinese students who scored higher in both psychological traditionality and modernity reported higher life satisfaction. Chinese students who scored higher in psychological traditionality reported experiencing more positive emotions and fewer negative emotions.

At the relational level of happiness, Chinese students who scored higher in psychological traditionality reported higher levels of life satisfaction within their families. Conversely, Chinese students who scored higher in psychological modernity reported having more positive relationships with others.

At the societal level of happiness, Chinese students who scored higher in psychological traditionality reported higher levels of social well-being.

Based on the above conclusions, it can be inferred that an individual’s happiness encompasses multiple levels. Therefore, in everyday life, individuals may benefit from not solely focusing on their happiness, emotions, and life satisfaction. By fostering positive relationships with family, friends, and others, individuals can find happiness by not only meeting their own needs but also by providing help and support to others and society. In turn, the well-being of others and the community becomes part of one’s own happiness.

## Figures and Tables

**Table 1 behavsci-14-00304-t001:** Descriptive statistical results of the participant’s demographic information.

Gender	*n*	Age (Years Old)	Subjective Social Class (Ten Thousand Yuan per Annum)	Objective Social Class
Males	217	25.02 (6.15)	21.27 (5.26)	6.01 (1.80)
Females	201	25.21 (6.17)	21.81 (6.44)	5.94 (1.69)
Total	418	25.11 (6.22)	21.54 (5.92)	5.97 (1.76)

Note: The values in the table were M (SD).

**Table 2 behavsci-14-00304-t002:** Descriptive statistics and correlation results of main variables.

*n* = 418		*M*	*SD*	*r* with PT	*r* with PM
Cultural Values	PT	187.37	21.77	/	/
	PM	183.05	4.19	0.346 (<0.001)	/
Individual-Level Happiness	SWLS	22.15	7.36	0.121 (0.016)	0.184 (<0.001)
	PANAS_positive	35.95	7.75	0.169 (<0.001)	0.142 (0.005)
	PANAS_negative	21.94	9.28	−0.87 (0.081)	0.076 (0.131)
Relational-Level Happiness	SWFS	22.48	7.79	0.067 (0.18)	0.116 (0.020)
	PWB_PR	54.36	7.75	0.115 (0.021)	0.220 (<0.001)
Societal-Level Happiness	SWBS	76.65	14.48	0.226 (0.001)	0.098 (0.051)

Note: The values in the table were *r* (*p*). *r* stood for the correlation coefficient, and *p* represented the significance value in correlation analysis. PT—psychological traditionality; PM—psychological modernity; SWLS—Satisfaction With Life Scale; PANAS—Positive and Negative Affect Schedule; SWFS—Satisfaction With Family Scale; PWB_PR—The dimension of “Positive Relationships” in the Psychological Well-Being Scale; SWBS—Social Well-Being Scale. Abbreviations are consistent in tables.

**Table 3 behavsci-14-00304-t003:** Hierarchical regression analysis results of PT and PM on individual-level happiness.

Variables	Individual-Level Happiness					
	Model 1			Model 2		
	(1) SWLS	(2) PANAS_P	(3) PANAS_N	(1) SWLS	(2) PANAS_P	(3) PANAS_N
PT				0.06 (<0.001)	0.1 (<0.001)	−0.14 (<0.001)
PM				0.03 (0.002)	0.02 (0.077)	0.07 (<0.001)
Gender	0.77 (0.212)	−1.73 (0.012)	−1.47 (0.114)	1.3 (0.037)	−1.48 (0.03)	0.19 (0.834)
Age	0.11 (0.029)	0.16 (0.005)	−0.27 (<0.001)	0.07 (0.166)	0.1 (0.047)	−0.26 (<0.001)
Subjective social class	2.23 (<0.001)	1.78 (<0.001)	−0.5 (0.068)	1.9 (<0.001)	1.41 (<0.001)	−0.49 (0.069)
Objective social class	<0.01 (0.274)	<0.01 (0.323)	<0.01 (0.975)	<0.01 (0.531)	<0.01 (0.67)	<0.01 (0.784)
Constant	5.97 (<0.001)	22.7 (<0.001)	32.53 (<0.001)	−8.19 (0.002)	3.89 (0.177)	45.88 (<0.001)
*n*	418	418	418	418	418	418
*R* ^2^	0.330	0.254	0.052	0.400	0.353	0.152
Adjusted *R*^2^	0.324	0.247	0.043	0.391	0.344	0.140
F, *p*	F(4, 417) = 50.89 *p* < 0.001	F(4, 417) = 35.19 *p* < 0.001	F(4, 417) = 5.68 *p* < 0.001	F(6, 417) = 45.89 *p* < 0.001	F(6, 417) = 37.37 *p* < 0.001	F(6, 417) = 12.28 *p* < 0.001

Note: The values in the table were β (*p*). β was the standardized beta coefficient (beta). *p* represented the significance value in regression analysis. Gender was treated as a categorical variable in this study. In the regression analysis, female participants were assigned a value of 1, while male participants were assigned a value of 0. The dichotomous coding method was employed for all other regression analyses in the entire study. Other coefficients in regression analysis (e.g., *SE* for standard error) can be found in the Appendix A for Table 3, Table 4 and Table 5.

**Table 4 behavsci-14-00304-t004:** Hierarchical regression analysis results of PT and PM on relational-level happiness.

Variables	Relational-Level Happiness			
	Model 1		Model 2	
	(1) SWFS	(2) PWB_PR	(1) SWFS	(2) PWB_PR
PT			0.06 (<0.001)	−0.01 (0.684)
PM			0.02 (0.047)	0.08 (<0.001)
Gender	0.58 (0.37)	−2.93 (<0.001)	0.9 (0.168)	−1.21 (0.107)
Age	0.12 (0.028)	−0.02 (0.783)	0.08 (0.125)	−0.06 (0.272)
Subjective social class	2.46 (<0.001)	0.95 (<0.001)	2.18 (<0.001)	0.53 (0.015)
Objective social class	0 (0.296)	0 (0.51)	0 (0.529)	0 (0.308)
Constant	5 (0.002)	51.22 (<0.001)	−7.95 (0.004)	41.45 (<0.001)
*n*	418	418	418	418
*R* ^2^	0.355	0.089	0.403	0.209
Adjust *R*^2^	0.349	0.081	0.395	0.198
F, *p*	F(4, 417) = 56.76 *p* < 0.001	F(4, 417) = 10.13 *p* < 0.001	F(6, 417) = 46.28 *p* < 0.001	F(6, 417) = 18.15 *p* < 0.001

**Table 5 behavsci-14-00304-t005:** Hierarchical regression analysis results of PT and PM on societal-level happiness.

Variables	Societal-Level Happiness	
	Model 1 (1) SWBS	Model 2 (1) SWBS
PT		0.24 (<0.001)
PM		0.02 (0.133)
Gender	−2.24 (0.074)	−1.97 (0.098)
Age	0.46 (<0.001)	0.35 (<0.001)
Subjective social class	3.52 (<0.001)	2.72 (<0.001)
Objective social class	0 (0.885)	0 (0.524)
Constant	46.22 (<0.001)	4.4 (0.383)
*n*	418	418
*R* ^2^	0.294	0.433
Adjust *R*^2^	0.288	0.424
F, *p*	F(4, 417) = 43.06 *p* < 0.001	F(6, 417) = 52.23 *p* < 0.001

## Data Availability

The data that support the findings of this study are available upon request. Due to confidentiality and privacy concerns, data can be downloaded after 31 May 2024 at https://www.scidb.cn/en/anonymous/TWJJejJx (accessed on 6 December 2023).

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
