# Peer review of "Psychological Traditionality and Modernity and Happiness: The Different Happiness Levels in Chinese Students"

_behavsci, 2024, doi:10.3390/bs14040304_

Round 1

Reviewer 1 Report (Previous Reviewer 2)

Comments and Suggestions for Authors

See attached file

Comments on the Quality of English Language

Extensive language editing is required.

Author Response

Thank you for your comments for my manuscript entitled “Psychological Traditionality and Modernity and Happiness: The Composition and Expanse of Happiness Circle in Chinese Students”(behavsci-2907184), I have already revised the manuscript according to your suggestion. Those comments are valuable and very helpful. We have read through comments carefully and have made corrections. In the revised manuscript, modifications made based on your comments are marked in red, changes made according to feedback from other reviewers are indicated in blue and green.

Please see the attachment for  a point-by-point response.

Reviewer 2 Report (Previous Reviewer 4)

Comments and Suggestions for Authors

After the revision, my concerns and queries have been clarified. One point still needs to be changed before publication. The table labels and notes are unclear or missing. A Reference should be made in the tables to which coefficients are reported (e.g. B and S.E.). It is also unclear what the note regarding Table 2 ("The values in the table were r (p)") means.

Author Response

Thank you for your comments for my manuscript entitled “Psychological Traditionality and Modernity and Happiness: The Composition and Expanse of Happiness Circle in Chinese Students”(behavsci-2907184), I have already revised the manuscript according to your suggestion. Those comments are valuable and very helpful. We have read through comments carefully and have made corrections. In the revised manuscript, modifications made based on your comments are marked in blue, changes made according to feedback from other reviewers are indicated in red and green.

Please see the attachment for  a point-by-point response.

Reviewer 3 Report (New Reviewer)

Comments and Suggestions for Authors

Thank you for the opportunity to review the manuscript entitled: “Psychological Traditionality and Modernity and Happiness: The Different Happiness Levels in Chinese Students”.

The issue of subjective well-being of students is of interest to society in general. It is convenient to study cultural differences and how they affect changes in values linked to modernity in order to prevent possible difficulties and make interventions, since the role of well-being in the mental health of individuals is key, and the prevalence of pathology in young people is currently of concern. I also find it very interesting to consider others´ well-being in the evaluation of self-happiness.

The authors' theoretical review of the subject seems complete and adequate. The references used are sufficient in number and very pertinent. Classic studies of referential authors are combined with current studies. Once the state of the question is finished, the hypotheses of the study are clearly formulated. In my opinion, they are very well founded. Only one aspect leaves me in doubt: why, in Hypothesis 1 (H1), do they expect that, at the individual level, individuals who scored higher in PT may report a higher frequency of positive emotional experiences and, conversely, individuals who scored higher in PM may report a higher frequency of negative emotional experiences.

Regarding the method, the authors specify the procedure for the conduction of the study. The participants give their consent and voluntarily agree to answer the questionnaires; however, they receive remuneration in return. According to the principles of the ethics committee of my university, participation in research must be disinterested, the participant cannot have any economic consideration; I do not know if it is correct to receive any benefit in other cultures for participating in research, this is strange to me.

The description of the variables could be completed: What are the cognitive attitudes, ideological concepts, value orientations, temperament traits, and behavioral intentions that are considered to be characteristic of traditional Chinese society as opposed to modern society? I found the answer in section 2.5. Measurement when reading the description of the Individual Multidimensional Traditionality and Modernity Scale (IMTMS), but I think that for those of us who do not know the reality of China in depth, this information should be more explicitly stated earlier, even in the theoretical framework. The definition of Individual-level happiness matches the one of subjective well-being since the model proposed by Diener (1984).

On the other hand, the questionnaires used are detailed, the number of participants is justified, the analyses carried out and the statistical package used are explained. The sample is also described in detail.

Regarding the results, given that it seems that both PT and PM are present in individuals, future studies could delve deeper into which dimensions of PT and PM are more related to life satisfaction and positive/negative affect. Some dimensions are incompatible with each other (e.g., believing that everything is determined or believing that my effort can change reality), although others may coexist. Perhaps a separate analysis could provide more information. On the other hand, it is not clear to me what explanation the authors give for the fact that subjects who score high in PM have more negative experiences. According to Diener's theory, the emotional balance of these individuals would be worse. The authors explain that emotional intensity probably compensates for frequency, but why do they think they experience more negative emotions?

I imagine the article is about to be formatted, but the title of Table 3 is loose, Table 4 is cut off. Also the note in Table 3 is confused with the text of the article. The margins of the tables fluctuate. Improving the tables formally will help to r consult the information better. There are also acronyms that do not appear in the footnotes and, although they have been explained in the previous text, they make it difficult to read the tables selectively.

In the references, a more thorough final review should be made. For example, numbers 10, 26, 35 or 59 have the authors' names in capital letters, e.g. ZENG. H.; GUO. S. P. There are journals that are not in italics (e.g., reference 15).

To conclude, and taking these aspects into account, in my opinion, the research analysed can be published and it is interesting, although it would improve with these small changes.

Minor revision.

Diener, E. (1984). Subjective Well-Being. Psychological Bulletin, 95(3), 542-575.

Author Response

Thank you for your comments for my manuscript entitled “Psychological Traditionality and Modernity and Happiness: The Composition and Expanse of Happiness Circle in Chinese Students”(behavsci-2907184), I have already revised the manuscript according to your suggestion. Those comments are valuable and very helpful. We have read through comments carefully and have made corrections. In the revised manuscript, modifications made based on your comments are marked in green, changes made according to feedback from other reviewers are indicated in red and blue.

Please see the attachment for  a point-by-point response.

This manuscript is a resubmission of an earlier submission. The following is a list of the peer review reports and author responses from that submission.

Round 1

Reviewer 1 Report

Comments and Suggestions for Authors

This paper would be far more interesting if its pool were not just limited to Chinese students. Limited to Chinese participants, it reads more as ideology in favor of traditional Chinese values, when in fact the findings “likely” reflect what it means to be a mature human being. The earlier differences between cultural notions of happiness likely have to do with researchers’ inconsistent definitions/measures of happiness (individualized vs collective notions of happiness), rather than differing feelings of happiness. It makes no sense that people inhabiting cultures that stress individuality would be more happy unless the researchers framed happiness that way. It looks like Diener’s research question was biased. Mature westerners, though maybe not college-age students, are no less likely to rate inner tranquility, harmonious co-existence and balance as more desirable than “hedonistic sensory pleasure.” I was pleased to see your crediting Chinese happiness to Confucian thinking and Buddhist avoidance of suffering. My non-Chinese students also aspire to the ideals of Confucius and the Buddha. If you want non-Chinese to read this paper, I would develop this section a bit more. By contrast, your description of non-Chinese evaluative models (PERMA, PANAS-scale and SWLS) is so cursory that I initially thought your point is that they are irrelevant for Chinese culture. I was thus deeply surprised that you ended up using them to evaluate your students’ individual happiness (psychological modernity) happiness. I don’t believe the authors describe the methods used to carry out these tests. 

I worry that the view of non-Chinese happiness is masculist, or at least influenced by “masculist” moralities/psychologies. By contrast, feminist ethics encourages collective approaches to care, generosity, justice, joy, etc., so feminist notions of happiness are no less collective than traditional Chinese notions. I would argue that a non-Chinese whose self is interdependent with others exhibits a “moral” (selfless) approach to happiness? 

After reading the text, I’m still not sure how to distinguish psychological traditionality from psychological modernity, though I understand that humility is considered increasingly less attractive, but I’m not sure why. To my lights, the authors not onlh insufficiently differentiate psychological traditionality from psychological modernity, but they fail to articulate the corresponding values. Plus, they use traditional virtues and cultural values interchangeably, yet values (unlike virtues) are technically individually-acted upon, even if people gain access to them via family or culture. There is a limit to the number of values each person can uphold and values often come into conflict with one another, which requires the individual to choose between values or balance them. I don’t think virtues are so conflictual. There needs to be a more robust distinction. The “happiness circle” is not unique to Chinese. Non-Chinese may have more tools for escaping it- move away, ignore family/friends who are sources of suffering, etc. Line 276 is confusing since it mentions 3 scales, but lists 6!

Comments on the Quality of English Language

This paper has scores of misspellings, missing words, etc. 

Reviewer 2 Report

Comments and Suggestions for Authors

See attached file

Comments on the Quality of English Language

Need significant editing

Reviewer 3 Report

Comments and Suggestions for Authors

The manuscript carries out interesting research for the scientific community. The analyzes included in the article can be used to analyze how happiness influences academic performance and academic persistence. The theoretical framework of the article serves to contextualize the objectives that have been set. Taking into consideration the relevance of interactive and social relationships in happiness, a comparative study between different Chinese university institutions could be interesting. From the research, doubts arise about the degree of interactive and social relationships depending on the origin of the students. One of the weaknesses of the research is the sample. The sample used can be considered low. The authors must justify this section from a theoretical point of view. The Conclusions section of the article should be expanded to consolidate the general vision of the article. The authors could propose some proposals that university institutions can offer to their students.

Reviewer 4 Report

Comments and Suggestions for Authors

Dear authors, 

please find enclosed my suggestions for the revision of your manuscript.

Round 2

Reviewer 1 Report

Comments and Suggestions for Authors

Unless psychological modernity is simply "not psychological traditionality," I still have no idea what psychological modernity is, let alone higher or lower instances; or how you assess whether an individual values PT or PM. It sounds like the researchers associate PM with "shopping/consumerism," which is inappropriate. Consumerism doesn't make anyone happy. It may yield temporary benefits.  

I think PM and PT not only need more precise definitions, but you must explain how you identify them in individuals. It would help to give examples of character types that exhibit PT vs PM. For example, it could be that Chinese females are more likely to value PT, while Chinese males are more likely to value PM. Or wealthier Chinese exhibit PM because they have more purchasing power. If this is the case, then this study sounds predictable. PM candidates lacking the resources to buy things would naturally be unhappy. At this point, you need a variable for wealth or individual purchasing power.

You assess gender in terms of happiness, but not in terms of whether people self-identify as PT or PM. Moreover, if people see themselves as having PT, then so long as they don't have to struggle to live a PT life, they should feel happy.

Maybe one has to be Chinese to understand/appreciate what the researchers are attempting to demonstrate with their data. 

Comments on the Quality of English Language

There are still plenty of mistakes but it is much better.

Reviewer 2 Report

Comments and Suggestions for Authors

Please see comments attached.

Comments on the Quality of English Language

Many concerns about this article remain on language.

Reviewer 4 Report

Comments and Suggestions for Authors

Dear authors,

thank you for the comprehensive revision of the paper in which almost all my comments were implemented.

I still miss a definition of psychological traditionality. Could you please elaborate on core aspects of psychological traditionality?

There are still some typos (e.g., "2.1 Participat"; 2.3 "Produce" - do you mean procedure?, 3.1 "Participant" instead of "Participants", Line 599 missing space after the dot behind the reported ANOVA).

Comments on the Quality of English Language

Please check again for typos and the usage of wrong terms (e.g. Produce instead of Procedure)